**Mediterranean Marine Heatwave 2023: Ecosystem and Fisheries impacts in Italian waters**
**Riccardo Martellucci R.**[1], Francesco Tiralongo F. [2],Sofia F. Darmaraki [3], Michela D'Alessandro [1], Giorgio
Mancinelli [4], Emanuele Mancini [4], Roberto Simonini [5], Milena Menna [1], Annunziata Pirro [1], Diego Borme [1], Rocco
Auriemma [1], Marco Graziano [1], Mauri Elena [1]
[1] National Institute of Oceanography and Applied Geophysics (Trieste, Italy)
[2] University of Catania (Catania, Italy)
[3] Foundation for Research and Technology Hellas (Heraklion, Greece)
[4] University of Salento (Lecce, Italy)
[5] University of Modena and Reggio Emilia (Modena, Italy)
Corresponding author: rmartellucci@ogs.it, sofia.darmaraki@dal.ca
**Abstract**
In 2023, the Mediterranean Sea experienced the longest recorded Marine Heatwave (MHW) in four decades, affecting
marine biodiversity, fisheries and coastal livelihoods. In this study, we assess the effects of this extreme event on the
spread of the invasive species *Callinectes sapidus* (Atlantic blue crab) and *Hermodice carunculata* (bearded fireworm)
along the Italian coasts. We focus on the coastal area of the Po Delta in the northern Adriatic and on the northern and
southern coasts of Sicily and investigate to what extent the increased seawater temperatures contribute to the increase
in the monthly biomass of these species. Considering that the spread of the Atlantic blue crab is responsible for
significant economic losses in the shellfish fishery and that the spread of the bearded fireworm poses a health risk to
artisanal fishermen, we also assess the socio-economic impact of this MHW by analysing fish market data and online
surveys. Finally, we discuss possible strategies to mitigate the spread and ecological impact of these invasive species.
We take into account the aggressive feeding behaviour of the fireworm and the thermophilic nature of the fireworm,
whose toxic antennae also pose a health risk to humans. Overall, the sustainability of marine ecosystems and coastal
communities in the Mediterranean requires robust interdisciplinary collaboration to address the challenges posed by
biological invasions and climate change in this region.

**1.Introduction**

The Mediterranean Sea, one of the most biodiverse marine ecosystems in the world (Coll et al., 2010), is currently
facing unprecedented challenges due to extreme temperature events caused by climate change, known as marine heat
waves (MHWs) (Darmaraki et al., 2019). In recent decades, the frequency, intensity and duration of these record-
breaking episodes have increased in the region, mainly due to the mean sea surface temperature (SST) warming trend
in the Mediterranean Sea, which ranges between 0.035–0.041 °C/year (EU Copernicus Marine Service Product,
2022a), almost twice as high as the corresponding global SST trend of $0.015 \pm 0.001$ °C/year (EU Copernicus Marine
Service Product, 2022b). This has significant implications for the region's biodiversity and economy, as the warming
trend and MHWs may facilitate the proliferation of invasive species (Joyce et al., 2024).
Species that pose a significant threat include the Atlantic blue crab, *Callinectes sapidus*, and the bearded fireworm,
*Hermodice carunculata*, which have attracted attention due to their rapid spread and negative impact on Italian
fisheries (e.g. Heilskov et al., 2006, Riera et al., 2014, Simonini et al., 2019, Righi et al., 2020, Bardelli et al., 2023,
Tiralongo et al., 2023). In particular, *C. sapidus*, which is native to the Atlantic, has rapidly colonised the Italian coasts
(Mancinelli et al., 2021). *C. sapidus* is characterised by its voracious predatory behaviour and opportunistic feeding
habits (Mancinelli et al., 2021) and has led to considerable economic losses in shellfish fisheries and enormous
challenges for native species (Clavero et al., 2022). Similarly, *H. carunculata*, a thermophilic polychaete, has spread
in Italian waters, affecting artisanal fisheries by both ruining the catch and posing health risks to human health
(Heilskov et al., 2006, Riera et al., 2014, Simonini et al., 2019, Righi et al., 2020, Tiralongo et al., 2023). In fact, *H.*
*carunculata* with its toxic setae can cause painful stings in humans, leading to burns and redness on physical contact
and posing a health risk for tourists in coastal areas and for fishermen, especially when cleaning nets (Tiralongo et al.,
2023). In addition, *H. carunculata* is an ecological disruptor as well as a direct threat to the well-being of coastal
communities. The resilience of both species to environmental stressors and the rapid spread of the population
emphasise the urgency of addressing the intensifying risks of climate change and bioinvaders with comprehensive
management strategies.
The year 2023 marked a turning point when average global air temperatures reached an unprecedented high
(Copernicus, 2024). The European continent experienced its second warmest year on record, with the Mediterranean
basin experiencing a series of extreme temperature events (Marullo et al., 2023). Of particular concern was the
occurrence of the longest recorded and one of the strongest surface MHWs of the last four decades, which persisted
in the northwestern Mediterranean from May 2022 to boreal spring 2023 (Marullo et al., 2023; Pirro et al., 2024
OSR8). At its peak in July 2022, this MHW covered almost the entire western Mediterranean basin, with maximum
daily SST anomalies reaching about 2.6 °C and 4.3 °C and anomalously warm conditions comparable to the 2003
summer MHW (Guinaldo et al., 2023). The long duration of the event was attributed to a combination of anomalously
low wind speeds, high solar radiation and weak vertical mixing in the ocean (Marullo et al., 2023). These warming-
related events had far-reaching impacts on marine life and coastal communities (He and Silliman, 2019).
The aim of this study is to investigate the spread and increase in abundance of *C. sapidus* and *H. carunculata* associated
with moderate and extreme warming in two coastal areas of Italy, particularly during the MHWs of 2022/2023.
Relevant socio-economic impacts were also assessed by analysing fish market data and responses to questionnaires
handed out to local fishermen and completed online, which addressed issues related to the bioinvasion of these species.
In addition, we discuss possible solutions to mitigate the invasion of these species.
**2. Methods**
*2.1 Study areas*
The study was conducted in two different regions within Italian waters: two adjacent lagoons in the Northern Adriatic
Sea (Canarin and Scardovari) and two coastal areas of Sicily (Figure 1).
2.1.1 The Po river delta
The two lagoons studied in the northern Adriatic are transitional and shallow water environments located in the Po
Delta and connected to the sea and various river branches (Figure 1b and 1c). As they are directly influenced by the
outflows of the Po River, these regions have highly dynamic hydro-morphological characteristics and are subject to
rapid changes due to biotic and abiotic forces (Maicu et al., 2018; Franzoi et al., 2023). Although the lagoons are
subject to various forms of anthropogenic pressure that have progressively altered their natural ecological
characteristics (Franzoi et al., 2023), the lagoons support several mussel and oyster farms, which represent the main
productive activities and vital economic resources at both local and regional level (Turolla et al., 2008; Donati &
Fabbro, 2010; Bordignon et al., 2020, Chiesa et al., 2025, Tiralongo et al. 2025). The Scardovari lagoon covers an
area of 32 km², has an average depth of 1.5 metres (Mistri et al., 2018) and is connected to the sea via two bays in the
north-east and south-west of the basin. The Canarin lagoon covers an area of about 6.4 km² in the southern part of the
study area and has an average depth of 0.9 metres. It is connected to the Adriatic Sea by a shallow, approximately 200
metres wide estuary with a maximum depth of 2.5 metres.
2.1.2 Sicily

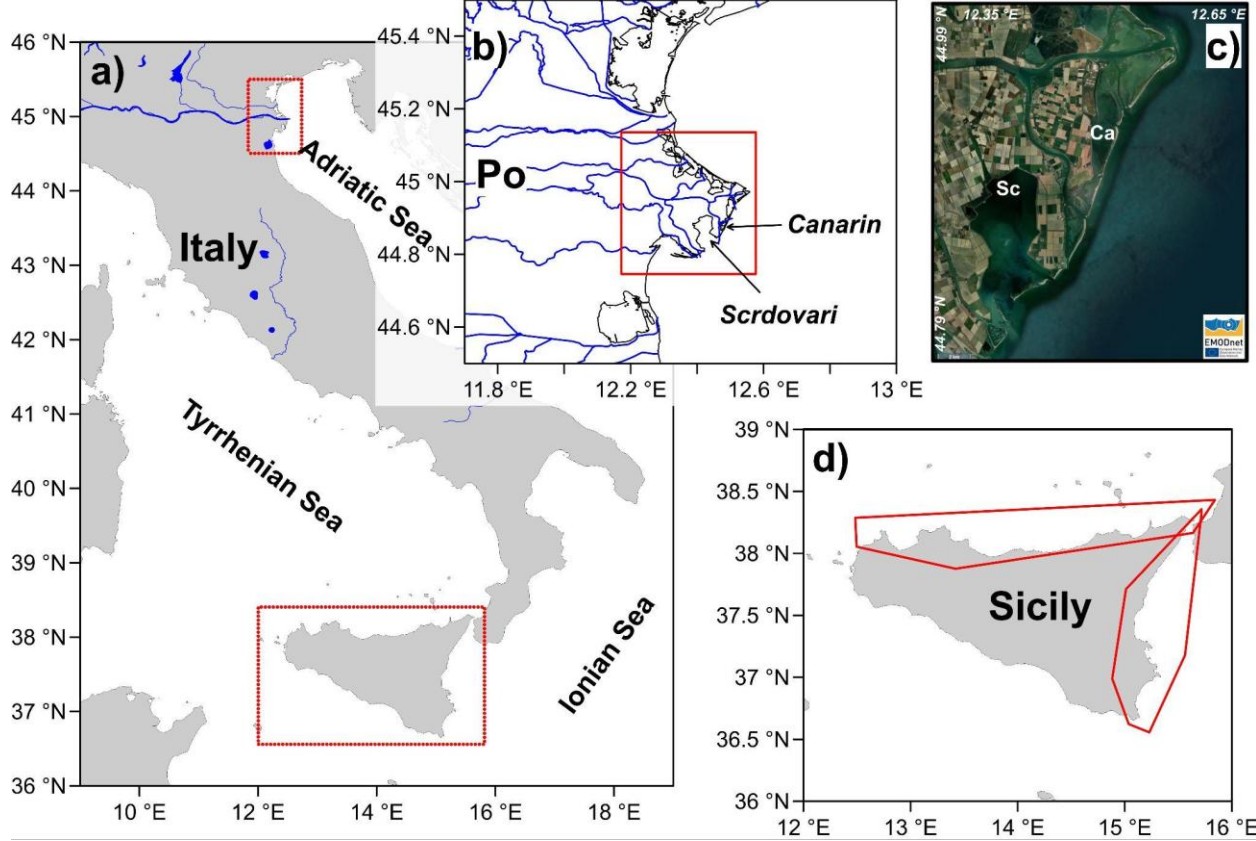


*Figure 1: Map of Italy (a) with the two study areas highlighted in red boxes. The map shows the main water basins in the region, with blue line indicating 10m Rivers and Lakes. (Retrieved from https://www.naturalearthdata.com). The Po river delta (b) and a detailed enlargement of the Scardovari (Sc) and Canarin (Ca) are highlighted in Panel C, downloaded from the EMODnet. Digital Bathymetry available at: https://portal.emodnet-bathymetry.eu/. The examined study (coastal) areas in Sicily island (d). For images of C. sapidus and H. carunculata, refer to Figures S4 and S5.*

The island of Sicily (Figure 1d) lies at the convergence of the eastern and western basins of the Mediterranean, which are influenced by both the relatively cooler Atlantic waters and the warmer Levantine waters. In particular, the eastern coast of Sicily (Ionian Sea) is significantly affected by quasidecadal reversals of the Northern Ionian Gyre driven by the mechanisms of the bimodal oscillation system in the Ionian Sea (Gačić et al., 2021; Menna et al., 2022). The distribution of water masses is altered by bringing in warm and salty Levantine water during the cyclonic phase (anti-clockwise) and transporting cooler Atlantic water during the anticyclonic phase (clockwise). This dynamic influences marine ecosystems and favours the occurrence of Levantine species during the cyclonic phase and vice versa (Civitarese et al., 2023). In comparison, the northern coast of Sicily (Tyrrhenian Sea) experiences less saline and relatively cooler Atlantic water flowing in through the Sardinian Channel (Vetrano et al., 2010), while the southern coast of Sicily is characterised by cold water due to semi-permanent upwelling (Raffa et al., 2017).

*2.2 Biological characteristics of the two species*

*2.2.1 Callinectes sapidus*

The Atlantic blue crab *C. sapidus* Rathbun, 1896 is a species native to the western coasts of the Atlantic Ocean and is naturally distributed from Nova Scotia to northern Argentina (Millikin and Williams, 1984). It was first recorded in Europe in 1901 and in the Mediterranean since 1947 (Mancinelli et al. 2021). The first record in Italian waters dates

back to 1949 from the lagoon of Venice and ballast water is considered the most likely cause (Nehring, 2011). In Mediterranean waters, *C. sapidus* is considered one of the 100 most invasive alien species (Zenetos et al., 2005; Katsanevakis et al., 2018; Tsirintanis et al., 2022) and is present in at least seven of the nine southern European marine ecoregions (Mancinelli et al., 2017a,b). In the last ten years, it has rapidly expanded its range to new ecosystems throughout the Mediterranean, such as the European Atlantic waters of Portugal, France, Belgium and Germany, but also in Italian waters (Tiralongo et al., 2021; Bardelli et al., 2023). This eurythermal and euryhaline species is a voracious predator characterised by aggressive behaviour, high fecundity, excellent swimming abilities and high fertility (Tsirintanis et al., 2022) and inhabits lagoons, estuaries and other coastal environments. *C. sapidus* has a complex, biphasic life cycle consisting of marine planktonic larvae (*zoea*) and benthic postlarvae (*megalopa*), with juveniles and adults living in estuaries, lagoons and other coastal habitats (Lipcius et al., 2007). In marine waters, this species lives mainly on soft substrates at depths between 1 and 90 metres. Their life cycle is very complex and includes different habitats depending on sex and ontogenetic stage: adults can reach a relatively large size, with a carapace up to 25 cm wide in males and 18 cm in females (Millikin and Williams, 1984), and reside in lagoons and estuaries where males settle and moult. After mating, the egg-laying females migrate to the sea where they lay their eggs. The young return to the transitional areas and, after rapid growth, reach sexual maturity in their second year of life (Millikin and Williams, 1984; Taylor et al., 2021). *C. sapidus* exhibits opportunistic feeding behaviour and feeds mainly on fish and invertebrates, especially bivalves and polychaetes, and may consume detritus and macrophytes when other food sources are scarce (Mancinelli et al., 2017; Tiralongo et al., 2024). Recent studies in the Po Delta have shown that *C. sapidus* significantly affects the aquaculture of *Ruditapes philippinarum* (Manila clam). Predation by *C. sapidus* resulted in mussel losses of up to 100 % in certain areas, with up to 56 % of mussel shells showing signs of predation and a complete absence of seeds in natural recruitment zones (Azzurro et al., 2025; Chiesa et al., 2025; Tiralongo et al., 2025). Although several studies have been conducted on this species, the impacts/interactions of *C. sapidus* on native species and Mediterranean aquatic ecosystems are still poorly understood and require further investigation (Mancinelli et al. 2017; Clavero et al. 2022).

### 2.2.2 *Hermodice carunculata*

The thermophilic amphinomid *H. carunculata* (Pallas, 1766), commonly known as the bearded fireworm, is a large predator/scavenger polychaeta found in warm and temperate areas of the Caribbean, Atlantic, Red Sea (Fishelson, 1971; Ahrens et al., 2013; Ramos & Schizas, 2023) and the Mediterranean Sea (Baird, 1868; Simonini et al., 2018; Toso et al., 2022, 2024). Although it is native to the Mediterranean region, it is also considered a highly invasive species due to its increasing spread. Previous studies indicate that its abundance across the Mediterranean has increased in recent years, likely due to warmer temperatures favouring its northward spread (Righi et al., 2020; Toso et al., 2022). This may have a detrimental effect on the region's ecosystems and associated species, as well as human health, as it is resilient to natural and anthropogenic stressors (Schulze et al., 2017) and may also become a carrier of new pathogens (Sussman et al., 2003; Schulze et al., 2017). Similarly, coastal and anthropogenic activities such as fishing (Figure S5) and bathing can also be affected (Celona & Comparetto, 2010; Cosentino & Giacobbe, 2011; Schulze et al., 2017; Simonini et al., 2018; Righi et al., 2020; Toso et al., 2020; Tiralongo et al., 2023). *H. carunculata* can grow to over 70 cm in length and reach a lifespan of 9 years (Simonini & Ferri, 2022). Its metamers are equipped with dorsal calcareous chaetae filled with a venom that is very effective against predators (Kicklighter and Hay, 2006; Schulze et al., 2017; Simonini et al., 2018, 2021; Righi et al., 2021, 2022). The presence of these defence mechanisms makes the polychaete highly resistant to predators, as none of the species identified in the Mediterranean region are able to effectively prey on it (Ladd & Shantz, 2016; Righi et al., 2021; Simonini et al., 2021). On the contrary, *H. carunculata* is a voracious predator of sessile and benthic invertebrates (Wolf and Nugues, 2013; Wolf et al., 2014; Jumars et al., 2015; Barroso et al., 2016; Schulze et al., 2017; Simonini et al., 2018; Righi et al., 2020), and its ability to regenerate favours its dispersal (Toso et al., 2024). In addition, this species is characterised by a remarkable dispersal ability, which is attributed to the production of planktotrophic and particularly long-lived larvae (Ahrens et al., 2013; Schulze et al., 2017; Toso et al., 2020). In Italian waters, *H. carunculata* is widespread on rocky substrates between 1 and 20 m (Righi et al., 2020; Simonini et al., 2021), but in some areas of the Mediterranean it reaches

greater depths and has also been observed in association with coralligenous and precoralligenous bioformations
(Fishelson, 1971; Righi et al., 2020).

Table 1: Products used in the present work. Complete references for the articles in Prod. 1, Prod. 3 and Prod. 6 are
reported in the bibliography.

| Ref. no. | Product name & type | Documentation |
|---|---|---|
| **Copernicus products** | | |
| 1 | Copernicus Marine SST_MED_SST_L4_REP_OBSERVATIONS_010_021 Mediterranean Sea - High Resolution L4 Sea Surface Temperature Reprocessed | Merchant et al., (2019) https://doi.org/10.48670/moi-00173 |
| 2 | Copernicus Marine MEDSEA_MULTIYEAR_PHY_006_004_E3R1 Mediterranean Sea Physics reanalysis | Escudier et al., (2021) Dataset: Escudier et al., (2020) https://doi.org/10.25423/CMCC/MEDSEA_MULTIYEAR_PHY_006_004_E3R1 |
| **Non Copernicus products** | | |
| 3 | Crab and clam fishery data | CONSORZIO COOPERATIVE PESCATORI DEL POLESINE Organizzazione di Produttori Soc. Coop. A r.l., Via della Sacca, 11 45018 Scardovari (RO) – ITALIA. P.IVA 00224140293 |
| 4 | Questionnaire Worms Out | Link: bit.ly/3L3TWUc https://www.facebook.com/MonitoraggioVermocane |
| 5 | Questionnaire Righi et al. 2020 | https://doi.org/10.12681/mms.23117 , |
| 6 | iNaturalist | https://www.inaturalist.org/ |

*2.3 Temperature Datasets*
To identify surface MHWs on the study areas we obtained daily SST data from the Mediterranean Sea SST Analysis
L4 product of the Copernicus Marine Service, covering the period 1982-2023 (Table 1, product ref. 1). This dataset
provides gap-free, optimally- interpolated, satellite-based estimates of SST with a resolution of 0.05°x0.05°. For the
analysis of subsurface temperatures in the areas of interest, daily vertical temperature profiles were obtained from the
Mediterranean Sea Physics Reanalysis dataset for the period from 1993to 2023 (Table 1, product ref. 2) with a spatial
resolution of 0.042° × 0.042°. MHWs are detected whenever the SST exceeds a daily, 40-year (1982-2023)
climatological threshold for at least 5 consecutive days, based on the identification framework proposed by Hobday
et al. (2016).
2.4 The crab and clam fishery data
To assess the impact of the spread of *C. sapidus* on the local fishing industry, we use data on the production of mussel
(*Ruditapes philippinarum*) provided by the Scardovari and Canarin Cooperative, which has been farming this species
in the Po Delta for years. The dataset contains monthly values for waste and sales of *C. sapidus*, representing the sum
of daily harvests by fishermen before reaching the market for fish sales (Table 1, ref. 3). Recent studies in the area
have shown that *C. sapidus* eats mussels, as evidenced by claw marks on the mussels. Compared to previous years,
mussel production in 2023 has decreased by 75 % in the Scardovari lagoon and 100 % in the Canarin lagoon (Azzurro
et al., 2025; Chiesa et al., 2025; Tiralongo et al., 2025), which is why we use discard data as an indicator of *C. sapidus*
biomass and damage to fisheries.
2.5 Questionnaire for *Hermodice carunculata*
In recent decades, citizen participation in the collection of data useful for science has increased thanks to numerous
awareness-raising initiatives (Turrini et al. 2018) and has already been recognised as a valuable resource for research,
biodiversity monitoring and conservation (Lopez et al., 2019; Toivonen et al., 2019). Although in some cases this
information lacks a solid scientific basis and needs to be validated by experts in the field, it offers the advantage that
it can be collected over wide geographical areas at low cost (Ballard et al., 2017; Tirelli et al., 2021; Sun et al., 2021).
For this reason, citizen science projects are currently on the rise in various areas, particularly as a tool for solving
environmental and conservation issues (Kullenberg et al., 2016; Turrini et al., 2018). For example, citizen participation
is widely used in projects and initiatives for the sighting of non-indigenous species, invasive and rare species such as
the *AlienFish* project (https://www.facebook.com/alienfish), *avvistAPP* (https://www.avvistapp.it/), *Monitoraggio*
*Vermocane* (https://www.facebook.com/MonitoraggioVermocane) and *iNaturalist* (https://www.inaturalist.org/).
To assess the impact of *H. carunculata* on human activities such as fishing and tourism, a questionnaire was developed
and distributed to fishermen as an online survey. The questionnaire, which builds on the observations of Righi et al.
(2020), was distributed to a large number of people in 2023, who provided a total of 151 responses. It was primarily
distributed via websites such as "*Monitoraggio del Vermocane*" and "*Fauna Marina Mediterranea*"
(https://www.facebook.com/groups/230601830399549) as well as via social media pages of various authors and also
forwarded to Italian diving centres. The questionnaire included 19 questions, of which four single-choice questions
focusing on the frequency of sightings, the abundance of specimens and the perception of the species as a potential
problem were analysed in this study. Most respondents reported having observed the fireworm while diving (74%),
spearfishing (10%) and snorkelling (15%), while no questions were asked about respondents' age or employment
status. Due to its distinctive morphology and bright colouration, *H. carunculata* is unlikely to be confused with other
species, as there are no comparable organisms in the Mediterranean. This study was carried out as part of the *Worms*
*Out* project funded by the National Institute of Oceanography and Applied Geophysics and the ECCSEL NatLab Italy
project (Table 1, ref. 5). In addition to the real-time data collected by marine users, the use of this questionnaire is
important to identify and assess the occurrence of this species. This is to address the current lack of scientific reports
on this species, which is probably due to the limited number of sampling efforts (e.g. Fraschetti et al., 2002;
Giangrande et al., 2003; Corriero et al., 2004; Mastrototaro et al., 2010). To address the scarcity of scientific reports,
we integrated observations from various sources such as the *iNaturalist* observations, the results of Righi et al. (2020)

and the most recent observations from the 2023 online survey. Integrating this information with scientific literature and observations is crucial for monitoring biological invasions and studying native invasive species (Azzurro et al., 2019; Giovos et al., 2019; Toivonen et al., 2019).

## 3. Results

### 3.1 Northern Adriatic

In the first half of 2023, mussel production in the area was around 400 tonnes per month and peaked in August (800 tonnes) before drastically declining from September to December (Figure 2a). In contrast, the sale of crabs began in the summer months, with the highest discard observed in August 2023 with a total of 300 tonnes. Towards the end of 2023, crab discards were comparable to sales (Figure 2b). Throughout 2023, the study area in the northern Adriatic was also characterised by particularly high SSTs, with six MHWs observed (Figure 2c and d). The first event occurred in March, lasted 5 days and had a moderate intensity, while the remaining events occurred during the summer and autumn seasons. The most intense MHW was observed at the end of August 2023 and was characterised by temperatures that were more than 3°C above normal and lasted 11 days (Table S1). The longest event was observed in October and lasted 36 days with a strong intensity (>2.6°C). The high temperatures of 2023 also affected the entire water column, with the strongest subsurface temperature anomalies (>4°C) observed between 4 m and 15 m depth during the summer season (Figure 2d). This indicates a significant increase in subsurface temperatures in this area, as typical deviations from the 1993–2016 mean are between 0.8°C and 1.2°C (Figure S1). Regarding the potential cumulative effects of temperature on the life cycle of *C. sapidus*, although the SST in 2023 was not below the threshold for hibernation (5°C), it exceeded the temperatures for reproductive activity (10.8°C) and larval development (19°C) by 300 and 170 days, respectively (dashed lines in Figure 2c).

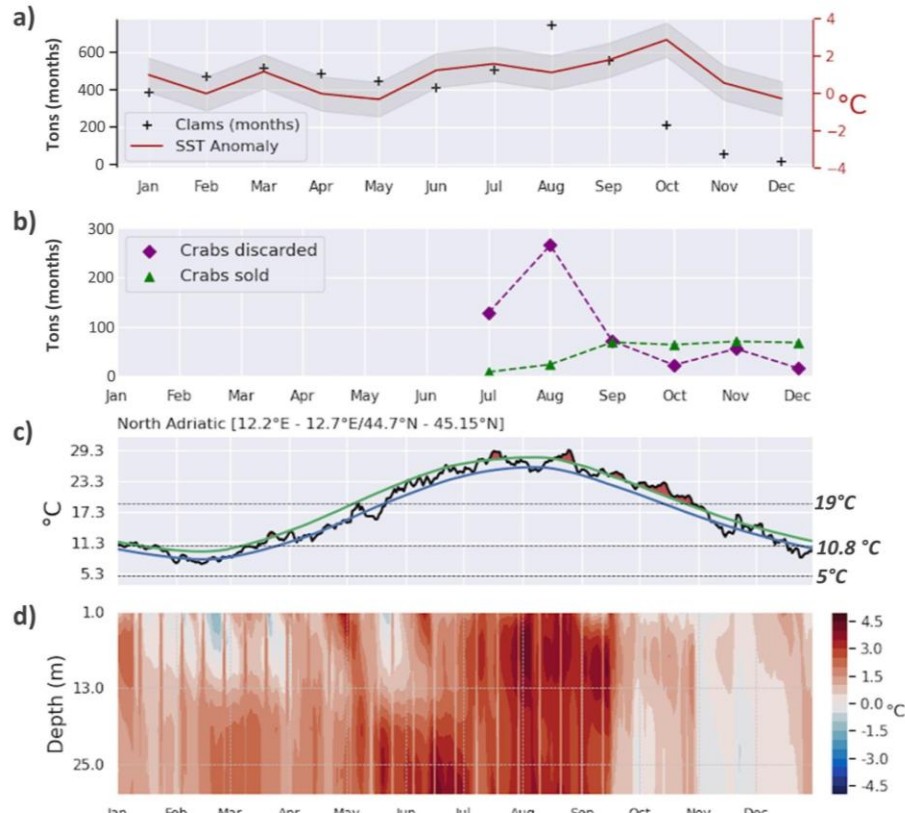

*Figure 2: Northern Adriatic study area: a) Time series of daily, spatially-averaged SST anomalies of 2023 relative to the period 1993-2016 (red line), and monthly evolution of sold clams (black cross). b) Monthly evolution of sold (green triangles) and discharged (purple diamonds) C. sapidus during 2023. c) Time series of daily, spatially-averaged SST during 2023 (black), smoothed SST climatology (blue) and 90th percentile threshold of SST (green) based on the 1982-2023 period. MHWs are indicated in red and identified using the Hobday et al. (2016) definition. The three dashed lines represent the temperature thresholds for winter dormancy (5°C), reproductive activity (10.8°C) and larval development (19°C) of C. sapidus. d) Vertical profile of temperature anomalies during 2023, relative to the period 1993-2016, spatially-averaged at each depth. Temperature data were obtained from Copernicus Marine Service (Table 1, product ref. 1, 2), Clams and Crab data were obtained from the Consorzio Cooperative Pescatori del Polesine (Table 1, product ref. 3)*

*3.2 Sicily*

In the two coastal regions of northern and eastern Sicily, the SST remained above the climatological values throughout 2023 (Figure 3b,c): Eastern Sicily experienced three MHWs that lasted about 60 days in total (Table S1). The most intense event occurred in July and lasted 21 days, the longest in October with a duration of 30 days. On the north coast of Sicily, there were slightly longer MHWs on average: the first event occurred in March, lasted 5 days and had an intensity of 1.4°C. The most intense MHW (>2.5°C) was observed between July and August with a duration of 25 days, while the longest event (49 days) occurred in autumn with an intensity of 1.6°C. Compared to the period 1993–2016, temperatures in the entire water column in both regions were around 1.2°C – 4°C warmer than normal, and even warmer on the north coast of Sicily. The upper 80 metres of the water column show the highest temperature anomalies (>2°C) throughout the year, especially in the autumn months (>2.5°C) in both areas. However, during some days in summer, the subsurface layers between 10 and 50 metres depth show negative temperature anomalies (down to –2°C) (Figure 3d,e), with temperatures dropping below 14 degrees at greater depths. Typically, subsurface temperatures deviate by about 0.8 – 2°C from the 1993–2016 mean, with the highest values observed in the upper 20–80 m depth between June–November (Figure S1). The progressive temperature increase of around 0.03°C/year observed on the north coast of Sicily seems to be consistent with an increasing trend in the records of *Hermodice carunculata* over the last 20 years, especially in 2007–2008, 2014–2015 and 2023, when the highest number of observations was recorded (Figure 3a). This increasing trend is also supported by the results of our proposed questionnaire, which shows a significant increase in the frequency of sightings in recent years compared to 2018, particularly during recreational activities. In particular, the results of the questionnaire show that the presence of this species is increasingly recognised as a problem, particularly as *H. carunculata* is observed in areas where it was not previously seen (Figure 3h and j). The data from the *Hermodice* questionnaire were compared with those from the iNaturalist platform (Table 1, ref. 6), and the two datasets showed similar observations, particularly in relation to the observed trend (Figure 2a).

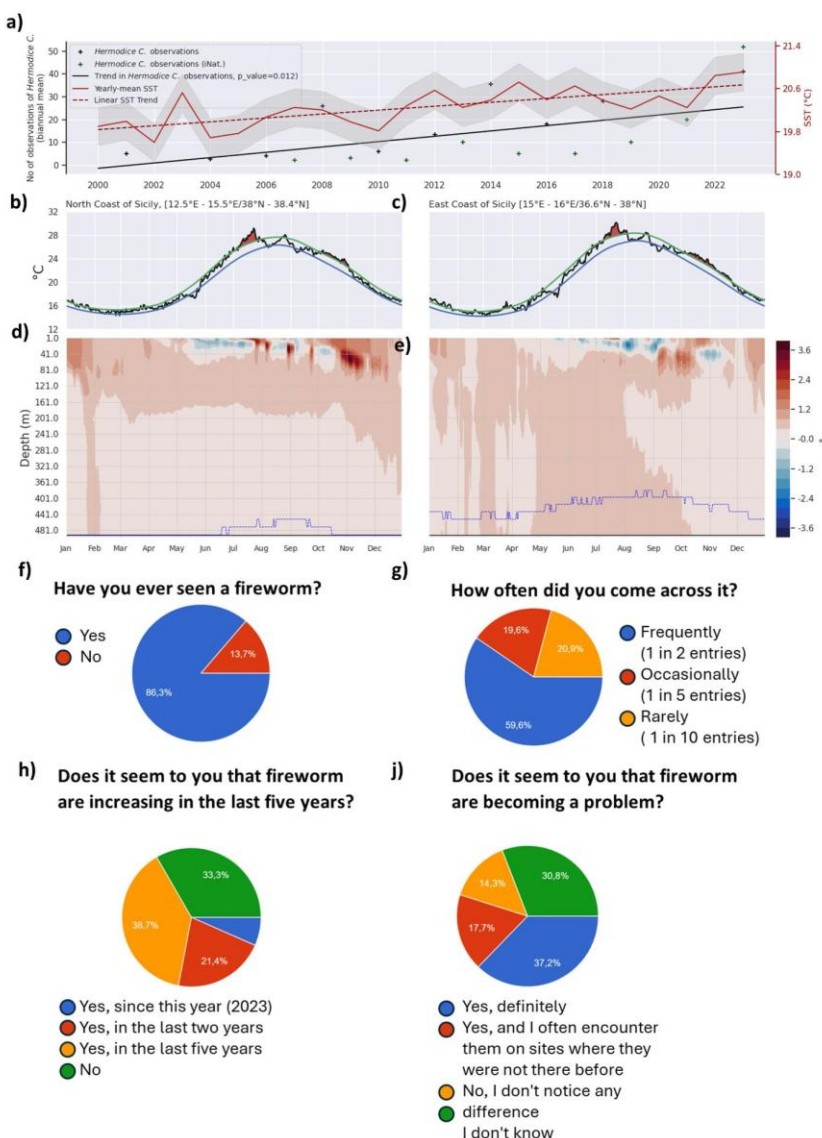

262

*Figure 3:* a) *Yearly-averaged time series and linear trend of SST (red), spatially-averaged over the Northern coast of Sicily and yearly records of H. caruncolata (black cross) with their linear trend, based on Righi et al. (2019) and our questionnaire for the period 2000-2023 (Table 1, product ref. 4 and 5), the green cross represent the biannual mean of H. caruncolata from iNaturalist. Daily and spatially-averaged SST time series during 2023 (black), smoothed SST climatology (blue) and 90th percentile threshold of SST (green) based on the 1982-2023 period, for the Northern (b) and Eastern (c) coast of Sicily. MHW are indicated in red and identified using the Hobday et al. (2016) definition. Vertical profile of spatially-averaged temperature anomalies during 2023, relative to the climatological period of 1993-2016 for the Northern (d) and Eastern (e) coast of Sicily. The climatological depth of 14 °C isotherm is displayed in blue dashed line whereas the depth of the 14 °C isotherm during 2023 in solid black. d) Main results of the Worms out questionnaire (Table 1, product ref. 4). Temperature data were obtained from Copernicus Marine Service (Table 1, product ref. 1, 2).*

274

## 4. Discussion

The two Mediterranean regions analysed in the present study experienced multiple, prolonged and strong surface MHWs throughout 2023, with temperature anomalies ranging between 1.6°C - 2.6°C on the eastern and northern coasts of Sicily and between 2-3°C on the northern Adriatic coast during the events. These events were associated with a general warming trend that was evident in increased monthly temperature anomalies occurring throughout the year, especially during summer in the northern Adriatic. Additionally, a trend of 0.03 °C per year was particularly evident in the sea surface temperature along the coast of northern Sicily during the period 2000–2023.Compared to the study areas in Sicily, the northern Adriatic showed a slightly stronger warming of the entire water column, which is probably due to its shallower depth. Also, lower than normal subsurface temperatures were observed in the upper 40 m of the two study areas in Sicily in summer and autumn. This is likely related to a stronger stratification of the upper ocean during these periods and a shift of the thermocline (see Figure S2,S3), resulting in cooler temperatures near the surface, similar to what Pirro et al. (2024) observed. Overall, a significant warming was observed in all study areas in 2023, which in turn may have led to an increase in the abundance of the two invasive species in both areas.

In the case of *Callinectes sapidus*, the higher temperatures may have triggered a positive feedback loop in the phenology of the larval and adult stages, ultimately leading to higher survival and reproduction rates and driving the population increase (Costlow 1967; Gencer, 2024). For marine invertebrates with complex life cycles, the effects of climate warming are particularly pronounced during critical stages (Libralato et al., 2015, Alter et al., 2024) such as larval development, reproductive activity, or winter dormancy (dashed line in Figure 2c). A similar mechanism has been proposed for other invasive crayfish such as *Hemigrapsus takanoi* as well as for other native brachyurans (Valdes et al., 1991; Anger, 2001; van den Brink et al., 2012; Oh and Lee, 2020). In this context, rising temperatures may have increased the invasiveness of the species in regions where established populations remained at low levels, as has been observed in the northern Adriatic Sea, where *C. sapidus* has been detected episodically since 1949 (Manfrin et al., 2016). Considering that in this species winter dormancy, adult reproduction and early life stages (egg maturation, zoea and megalopa development) are strongly regulated by temperature minima (Brylawsky and Miller 2006; Rogers et al., 2022; Schneider et al., 2024), an increase in water temperatures may have additionally accelerated metamorphosis from zoea to megalopa, reducing predation risk, promoting survival and ultimately population abundance. On the other hand, a further increase in temperature above 26°C may have an opposite effect by reducing the number of moults per larva and causing a reduction in blue crab growth (Gencer, 2024).

An increase in water temperatures also accelerates development and growth rates in *Hermodice carunculata* (Libralato et al., 2015, Alter et al., 2024) and several studies identify in water temperature as a decisive factor in the spread of the species (Righi et al, 2020, Tiralongo et al., 2023, Toso et al., 2024) by influencing the species range shifts and facilitating its establishment and spread (Stachowicz et al., 2002; Samperio-Ramos et al., 2015). Our questionnaires and iNaturalist observations confirm this spread and show a significant increase in sightings over the last five years. Our results are thus consistent with the documented increase in the presence of the species and its bathymetric extent over the last two decades, which coincides with a general increase in water temperatures along the coasts of Sicily (Pisano et al., 2020; Righi et al., 2020, Tiralongo et al., 2023; Kubin et al., 2024), which has been particularly evident since 2023 (Figure 3a). Given the ongoing warming trend in the Mediterranean, it is likely that this thermophilic species will continue to spread along the north-western Mediterranean coast, where it was frequently observed in shallow water conditions, especially in the summer months (Schulze et al., 2017; Encarnação et al., 2019; Righi et al. 2020). Noticeably, both the remarkable increase in the monthly biomass of *C. sapidus* in the coastal areas of the northern Adriatic and the annual records of *H. carunculata* on the coasts of Sicily share a number of common implications. In particular, we show that the proliferation of *C. sapidus* in the Po river delta significantly disrupted mussel production in 2023 (Figure S4) and that, under current conditions, no resurgence of mussel populations is expected in 2024 (Chiesa et al., 2025). The dramatic impact on the mussel fishery is exacerbated by the cost of removing the crabs discarded by the fishery (Figure S4). This expansion is also expected to disrupt benthic habitats, altering community composition and leading to remarkable changes at the level of the entire food web, as observed in

the Ebro River Delta (Clavero et al., 2022) and recently in the Po River Delta in the Goro Lagoon (Gavioli et al., 2025).

Similarly, *H. carunculata* poses a threat to both marine biodiversity and the economic stability of local fisheries, as has been observed for other annelids (Berke et al., 2010; Pires et al., 2015). Although comprehensive information on the impact of the species has only recently been presented (Tiralongo et al., 2023), the fireworm is known to cause both direct and indirect damage to fisheries. Direct damage includes the severing of secondary lines attached to hooks, either by the worm's teeth or by hiding among the rocks after consuming the bait (Tiralongo et al., 2023). To minimise this impact on the target species, fishermen reduce the soaking time of the fishing gear, which leads to a decrease in catch rates (Simonini et al., 2021; Tiralongo, 2020). The scavenging activities of *H. carunculata* not only affect the efficiency of fisheries, but also impact the fishing industry by damaging fish catches and reducing the market value of fish. The economic impact is estimated at around € 7.32 per kilogramme of damaged fish, resulting in significant annual losses given the total weight of commercially valuable catches (Tiralongo et al., 2023). The continued temperature-induced spread of *H. carunculata*, as observed in our results and documented by Righi et al. (2020) and Tiralongo et al. (2023), emphasises the urgent need for effective mitigation strategies to address the impacts caused by climate change on fisheries, tourism and the coastal economy.

*Implication for human life and solutions for stakeholder*

The ongoing invasion of *C. sapidus* provides an opportunity to evaluate strategies and measures to contain this spread and mitigate its ecological impact, even in freshwater ecosystems (Scalici et al., 2022; Tiralongo et al., 2024b; Bedmar et al., 2024). As the economic value of the species is already internationally recognised, this species can be used for both food and other purposes. Several studies have shown that overharvesting plays an important role in the control of invasive species (Mancinelli et al., 2017), as exemplified by the commercial harvest of *C. sapidus*, which supports a significant fishery on the coasts of the USA (Hines, 2007, Kennedy et al., 2007; Bunnell et al., 2010); accordingly, control measures should target similar commercialisation strategies. Despite its introduction to the Italian market, the consumption of *C. sapidus* is not yet widespread, complicating efforts to eradicate it (Azzurro et al., 2024). Effective management strategies should therefore include the cultural integration of this species. For example, the Italian government has promoted *C. sapidus* by showcasing it at the 2024 G7 summit and distributing online promotional content. Expanding outreach efforts through targeted events and educational campaigns could further boost consumption of this species, similar to what has been done successfully with other species. Furthermore, the extraction of chitosan and astaxanthin from crab shells could support the ongoing shellfish market while providing valuable compounds with multiple applications in pharmaceutical, biomedical, cosmetic, agricultural and biotechnological fields (Ambati et al., 2014; see also Demir et al., 2016, Baron et al., 2017 for recent examples on *C. sapidus*).

Similarly, the increasing frequency of *H. carunculata* sightings emphasises the need for effective management strategies to contain its spread. As an efficient scavenger, predatory generalist and opportunistic consumer that can also feed on carrion, *H. carunculata* has also been found in large numbers under aquaculture net cages and in places with high anthropogenic pressure and organic enrichment, such as artisanal fishing harbours (Heilskov et al., 2006, Riera et al., 2014; Righi et al., 2020). Due to its ability to tolerate captivity, *H. carunculata* offers potential applications for the disposal of waste from the production and processing of marine products. Recent biorefinery research is investigating the use of *H. carunculata* in the processing of molluscan waste, particularly expired mussels from retail outlets, to recover and valorise the shells. Preliminary results show that *H. carunculata* consumes mussel meat at high rates and leaves the shells almost completely clean (Simonini et al., 2024). Given its ability to maintain them at high densities without substrate, the species could prove useful for the valorisation of waste shells, with clean shells serving as a source of "green" calcium carbonate (Seesanong et al., 2023). In addition, experiments could be conducted to determine the effectiveness of this species when used in IMTA (Integrated Multitrophic Aquaculture) systems to determine its performance as a bioremediator organism (Giangrande et al., 2020). The development of practical

applications for this invasive species could also support the elimination of areas where *H. carunculata* becomes a pest
(Simonini et al. 2024).
Thus, the costs of managing and controlling invaded habitats may ultimately yield gains for the local population, while
greatly reducing the impact of the invader and even enhancing the ecosystem goods and services provided by coastal
habitats. Collaborative efforts are essential to formulate adaptation measures that protect both marine ecosystems and
the livelihoods of Mediterranean coastal communities. Through interdisciplinary collaboration and proactive
management strategies, it is possible to mitigate the negative impacts of climate change and the spread of invasive
species, thus ensuring the long-term sustainability of the Mediterranean marine environment and the well-being of
coastal communities.
**5. Conclusions**
Overall, global warming and biological invasions in marine ecosystems are so far recognised to be closely linked,
although the extent of their interactions and the role of climate change as a driving force remain controversial, as they
vary along the invasion process and are influenced by species-specific responses to warming (Blackburn et al., 2011;
Katsanevakis et al., 2014, Joyce et al., 2024). These responses affect the distribution, demography and life histories
of invasive species. Currently, there is an urgent need for additional studies investigating the relationship between
climate warming and the phenology of bioinvasives to elucidate the indirect effects that may occur on both the
distribution and abundance of the latter. The present study, focusing on *Callinectes sapidus* and *Hermodice*
*carunculata*, is one of the first attempts to address this important issue in the Mediterranean. However, temperature
alone may not be the only determinant of increases in invasive species abundance: for example, for *H. carunculata*,
Simonini et al. (2021) suggest that fishing practices may facilitate the spread, establishment and survival of the species,
with fish waste and other organic debris resulting from the cleaning of nets potentially providing increased trophic
resource availability. In addition, releasing pregnant *C. sapidus* females after capture can significantly increase the
number of larvae and increase populations (Hines et al., 2008). In the Po Delta, this practise has become common
among fishing companies given the low value of egg-laying females in local fish markets (Tiralongo, personal
observation; see also Figure S4). These examples illustrate how inadequate management practises can inadvertently
favour the spread of invasive species, and ultimately highlight the need for integrated control and mitigation strategies
that take into account the diverse range of factors that can actually contribute to the success of a bioinvasion.
**Acknowledgements**
Darmaraki S. acknowledge the financial support from the Hellenic Foundation for Research and Innovation (HFRI)
under the 3rd Call of "Research projects to support Post-Doctoral Researchers" scheme, Project number 7077,
acronym TexMed. This work was partly funded by the ECCSEL-NatLab Italy project, made by the extraordinary
MIUR contribution (FOE) for the participation of Italy in the activities related to the international ECCSEL
infrastructure (European Carbon Dioxide Capture and Storage Laboratory Infrastructure), with the aim of developing
and maintaining of the natural laboratories of Panarea (Aeolian Islands) and Latera (Lazio).

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
