# Peer review of "Mediterranean Marine Heatwave 2023: Ecosystem and Fisheries impacts in Italian waters"

_State of the Planet, 2024_

## Author Response (AR1)

Dear Reviewers,

Thank you for your valuable feedback and suggestions. Following up on our initial response to the review, we have further revised the manuscript to accommodate your recommendations while ensuring consistency with the structure of the Special Issue. Specifically, we have changed the title, modified the abstract, and reorganized the discussion section for better integration of the topics.

Additionally, we have included more details regarding clam production and incorporated an additional dataset for *Hermodice carunculata* from ìNaturalist.

All modifications made in the manuscript are highlighted in red for your convenience.

We sincerely appreciate your time and effort in reviewing our work.

Specific response

REV1

1. Title Scope: The heading appears too broad, as the study primarily focuses on a limited number of coastal regions in Italy and only examines two species of bioinvaders. I recommend refining the title for better clarity and specificity regarding the study's focus.

Thank you for your comment the title changed as follows:

*Mediterranean Marine Heatwave 2023: Ecosystem and Fisheries impacts in Italian waters*

2. Data Correlation: There is a lack of direct correlated data between temperature and the abundances or catches of the two species (Callinectes crab and Hermodice worm). The data presented seem to originate from only one location, which limits the study's robustness.

In this paper it was chosen to show trends in temperature and abundance of the species, avoiding the inclusion of direct correlations between the two variables, this is to show exclusively that in the last year the increase involved both temperature and abundance of these species.

I do not consider the comment about a limited number of coastal regions to be correct, let me explain further, regarding blue crab, this is the first time that a proliferation of the species has been observed in the Venetian lagoon, which is an important economic source for the clam trade. While with regard to Hermodice, the region of Sicily represents the first one conditioned by the proliferation of the organism and the one on which the questionnaire has been massively applied.

Moreover, these regions are the ones where data were collected for this study. For these reasons it was chosen to focus the work on these areas.

3. Sampling Methodology: The manuscript does not provide sufficient information about the sampling effort for the Hermodice data. Details such as the number of participants involved in the study, their age range, and work status are necessary for transparency and to assess the reliability of the findings.

Thanks for the comment we include new text to better specify the sampling methodology. The text changed as follows:

*Over the last decades, citizen participation in data collection useful for science has increased, thanks to numerous awareness-raising initiatives (Turrini et al. 2018) and has already been recognized as a valuable resource for research, biodiversity monitoring, and conservation (Lopez et al., 2019; Toivonen et al., 2019). While, in some cases, this information lacks a solid scientific basis, requiring validation by experts in the field, it offers the advantage of being gathered over broad geographical areas at a low cost (Ballard et al., 2017; Tirelli et al., 2021; Sun et al., 2021). For this reason, citizen science projects are currently increasing in several fields, especially as a tool to address environmental and conservation issues (Kullenberg et al., 2016; Turrini et al., 2018). For instance, citizen involvement is widely used in projects and initiatives related to the sighting of non-indigenous species, invasive and uncommon species, such as AlienFish project (https://www.facebook.com/alienfish), avvistAPP (https://www.avvistapp.it/), Monitoraggio Vermocane (https://www.facebook.com/MonitoraggioVermocane) and ìNaturalist (https://www.inaturalist.org/).*

*To assess the impact of H. carunculata on human activities, such as fishing and tourism, a questionnaire was developed and administered to fishermen and distributed as an online survey. The questionnaire built upon the observations by Righi et al. (2020) was administered in 2023 to a diverse group of individuals, yielding a total of 151 responses. Distributed primarily through websites, such as the "Monitoraggio del Vermocane", and the "Fauna Marina Mediterranea" (https://www.facebook.com/groups/230601830399549) and via social media pages of various authors, it was also shared with Italian scuba diving centers. The questionnaire comprised 19 questions, of which four single-choice questions, focusing on the frequency of sightings, the abundance of specimens, and perceptions of the species as a potential issue, were analyzed in this study. Most respondents reported observing the fireworm during scuba diving (74%), spearfishing (10%), and snorkeling (15%) activities, while no questions about the respondents' age or employment status were included. Due to its distinct morphology and vibrant coloration, H. carunculata is unlikely to be mistaken for other species, as no comparable organisms exist in the Mediterranean. This survey was conducted as part of the Worms Out project, funded by the National Institute of Oceanography and Applied Geophysics and the ECCSEL NatLab Italy project (Table 1, ref. 5). In addition to real-time data collected from maritime users, the use of this questionnaire is important for identifying and assessing the presence of this species, addressing the current lack of scientific reports on the species, likely due to sampling limitations (e.g., Fraschetti et al., 2002; Giangrande et al., 2003; Corriero et al., 2004; Mastrototaro et al., 2010). To address the scarcity of scientific reports, we integrated observations from different sources such as the ìNaturalist observations, findings of Righi et al. (2020) and recent observations from the online survey 2023. Integrating this information with scientific literature and observations, is crucial for monitoring biological invasions and studying native invader species (Azzurro et al., 2019; Giovos et al., 2019; Toivonen et al., 2019).*

4. Clam Production Data: The study lacks information on clam production, including species names and potential interactions with the studied crab (e.g., predation). Supporting references should be included to strengthen this discussion.

Thanks for the comment, we include this part in the manuscript, the text changed as follows:

*C. sapidus exhibits opportunistic feeding behavior, primarily preying on fish and invertebrates, especially bivalves and polychaetas, and can consume detritus and macrophytes when other food sources are*

*scarce (Mancinelli et al., 2017; Tiralongo et al., 2024). Recent studies conducted in the Po delta revealed that C. sapidus significantly affected the aquaculture of Ruditapes philippinarum (Manila clam). Predation by C. sapidus resulted in clam losses of up to 100% in certain areas, with up to 56% of clam shells showing signs of predation and a complete absence of seeds in natural recruitment zones (Azzurro et al., 2025; Chiesa et al., 2025; Tiralongo et al., 2025).*

5. Integration of Arguments: The discussion and conclusion sections lack a cohesive flow of arguments. It is crucial to integrate the data on the crab, worm, and temperature more effectively. Several assumptions made in these sections do not seem adequately supported by the data or existing literature.

Thank you for your insightful feedback. While we initially disagreed with the comment regarding the lack of integration between the data on the blue crab, Hermodice, and temperature, we have taken your suggestion into account and made several improvements.

In response to your comment, we have worked to better integrate these discussions and ensure the assumptions made are more clearly supported by the available data and literature. We have refined the flow of arguments in the discussion section to strengthen the connection between the temperature increase and the species' proliferation. We believe these revisions address your concerns while maintaining the original observations and insights of the study. Thank you again for your valuable input.

The discussion section changed as follows:

[revised manuscript text omitted]

6. Conclusions Specificity: The final paragraph of the conclusions is overly generic. More details regarding the studied regions and proposed management strategies would enhance the relevance and applicability of the findings.

*In this case, although what is written in the commentary is absolutely right, I would like to point out that the ocean state report has very specific characteristics that make it differ from a research paper.*

*In this case, the chapter is on the topic of problems and solutions for stakeholders, i.e., it has a very specific slant that we the authors have tried to adhere to, first by showing what is the problem, which is established and serious, second we have described some good practices aimed at avoiding problems related to the proliferation of these species.*

*The link between temperature increase and species proliferation is an extremely important research topic, although much laboratory research has shown direct relationships between such proliferation and temperature increase, in the natural environment this is extremely more complex to assess, but more importantly it is not the scientific focus of the paper.*

*Thaking into account your comment we changed the last chapter as follows:*

*sapidus invasion presents an opportunity to evaluate strategies and measures to contain this dispersion and mitigate ecological impacts. As their economic value has already been recognized internationally, this species can be exploited for both food and non-food uses. Several studies have demonstrated that overharvesting plays an important role in the control of invasive species (Mancinelli et al., 2017), as exemplified by the commercial harvest of C. sapidus, which supports a major fishery along the coasts of US (Hines, 2007, Kennedy et al., 2007; Bunnell et al., 2010); accordingly, control policies should aim at similar marketing strategies. Despite its introduction to the Italian market, C. sapidus is not yet widely consumed, complicating efforts to eradicate it. Effective management strategies should therefore include cultural incorporation of this species. For example, the Italian government has promoted C. sapidus by presenting it at the 2024 G7 summit and by disseminating promotional online content. Expanding public outreach through targeted events and education campaigns could further boost its consumption, akin to successful models used for other species. Additionally, the extraction of chitosan and astaxanthin from crab shells may support the ongoing shellfish market, while providing valuable compounds with diverse applications in pharmaceutical, biomedical, cosmetic, agricultural, and biotechnological fields (Ambati et al., 2014; see also Demir et al., 2016, Baron et al., 2017 for recent examples on C. sapidus).*

*Similarly, the increasing frequency of H. carunculata sightings highlights the need for effective management strategies to mitigate its proliferation. As an efficient scavenger, predatory generalist and opportunistic consumer that can also feed on carrions, H. carunculata has been also found in high abundance beneath aquaculture net cages and at sites with high anthropogenic pressure and organic*

*enrichment, such as artisanal fishery ports (Heilskov et al., 2006, Riera et al., 2014; Righi et al., 2020). Due to its ability to tolerate captivity, H. carunculata presents potential applications for the disposal of waste from the production and processing of marine products. Recent biorefinery research explores its use in processing mollusc waste, specifically expired mussels from retailers, for shell recovery and valorization . Preliminary findings show that H. carunculata consumes mussel meat at high rates, leaving the shells almost completely clean (Simonini et al., 2024). Given its ability to maintain them at high densities without substrate, the species could prove useful for valorizing waste shells, with clean shells serving as a source of "green" calcium carbonate (Seesanong et al., 2023). Developing practical applications for this invasive species could also support removal interventions from areas where H. carunculata is becoming a pest (Simonini et al. 2024).*

*Thus, management and control costs in invaded habitats may ultimately yield profits for local populations, while the effects of the invader may be greatly reduced, even enhancing the ecosystem goods and services provided by coastal habitats. Collaborative efforts are essential for formulating adaptive measures to safeguard both marine ecosystems and the livelihoods of communities along the Mediterranean coasts. Through interdisciplinary cooperation and proactive management strategies, it is possible to mitigate the adverse effects of climate change and invasive species proliferation, ensuring the long-term sustainability of Mediterranean marine environments and the well-being of coastal communities.*

**Rev2**

Thank you for the comments. The suggestions have been incorporated into the text.

**Title:** Record-breaking 2023 temperatures in the Mediterranean Sea, proliferation of bioinvaders, and impacts on fisheries: a chain reaction?

The second part: " impacts on fisheries: a chain reaction?"is discussed extensively in the discussion. However, it is only marginally part of the results (questionnaire), and as such the title is misleading.

Thank you for your comment, the title changed as follows:

*Mediterranean Marine Heatwave 2023: Ecosystem and Fisheries impacts in Italian waters*

**Abstract**

Lines 30-36 discuss potential management measures to control the population of the two invaders, but these are based on literature only, not results of this study. I suggest removing the last paragraph from the abstract.

Thanks for the comment, the abstract was modified as follows:

*In 2023, the Mediterranean Sea experienced the longest recorded Marine Heatwave (MHW) in four decades, affecting marine biodiversity, fisheries and coastal livelihoods. In this study, we assess the*

*effects of this extreme event on the proliferation of the invasive species Callinectes sapidus (Atlantic blue crab) and Hermodice carunculata (bearded fireworm), along Italian coasts. Focusing on the coastal area of Po River Delta in the northern Adriatic Sea and on the Northern and Southern Sicily coast, we explore the possible contribution of elevated seawater temperatures in increasing the monthly biomass of these species. Given that, the expansion of the Atlantic blue crab is responsible for substantial economic losses in bivalve fisheries and the proliferation of bearded fireworm poses health risks to artisanal fishers, we further assess the socio-economic implications of this MHW, through an analysis of fish market data and online survey responses. Finally, we discuss potential mitigation strategies to manage the spread and ecological impact of these invasive species, considering the aggressive feeding behavior of the former and the thermophilic nature of the latter, whose venomous setae also poses health risks to humans. Overall, the sustainability of Mediterranean marine ecosystems and coastal communities requires robust interdisciplinary collaboration to address the challenges posed by biological invasions and climate change in the region.*

**Introduction**

Thanks for the comments all typo error have been corrected

Line 57: Bardelli et al., 2023. This source refers to the crab only, not the worm,

Sorry for this we changed the text as follows:

*Among the species that pose a significant threat are the Atlantic blue crab, Callinectes sapidus, and the bearded fireworm, Hermodice carunculata, which have gained attention, as a result of their rapid expansion and adverse impacts on Italian fisheries (e.g., Heilskov et al., 2006, Riera et al., 2014, Simonini et al., 2019, Righi et al., 2020, Bardelli et al., 2023, Tiralongo et al., 2023).*

Line 70-71: This event had far-reaching effects on marine life and coastal communities. Please give reference(s) to it

Thanks for your suggestion, the text was modified as follows:

The long duration of the event was attributed to a combination of anomalously low wind speeds, high insolation, and weak vertical mixing in the ocean (Marullo et al., 2023). These worming related events had far-reaching effects on marine life and coastal communities (He and Silliman, 2019).

**Methods**

Lines 80-81: The study was conducted in two different regions within Italian waters: two adjacent lagoons in the Northern Adriatic 80 Sea (Canarin and Scardovari) and two coastal areas of Sicily.

Please move the sentence under Study areas (line 78)

Thanks for this we moved the first sentence under the paragraph 2.1

Figure 1. The images of C. sapidus and H. carunculata are redundant and rather misleading. C corresponds to the area not to the H. carunculata image.

I apologize for this, Figure 1 has been redone to better highlight the two study areas. The images of the animals have been removed, with references made to the images in the supplementary material.

Line 124:   Streftaris and Zenetos, 2006 is an old source. I recomment using Tsirintanis et al. 2022.

Tsirintanis K, Azzurro E, Crocetta F, Dimiza M, Froglia C, Gerovasileiou V, Langeneck J, Mancinelli G, Rosso A, Stern N, Triantaphyllou M, Tsiamis K, Turon X, Verlaque M, Zenetos A, Katsanevakis S (2022) Bioinvasion impacts on biodiversity, ecosystem services, and human health in the Mediterranean Sea. Aquatic Invasions 17(3): 308–352, https://doi.org/10.3391/ai. 2022.17.3.01

Thanks for this, we changed the reference with the newest ones.

Line 134: Hermodice carunculata

Thank you for your feedback. We have made the necessary corrections, including changing all the common species names to their Latin names

The way the lines 135-138 are written it appears that H. carunculata is an alien species. In fact. it is considered a Mediterranean species, not an introduced one. After all, all Mediterranean Sea biota are either of Atlantic or Indo-Pacific origin. It is invasive but not alien.

Absolutely agree with your comment, and thank you for the clarification, the authors are aware that this is not an alien species and the text was corrected as follows:

*The thermophilic amphinomid H. carunculata (Pallas, 1766), commonly known as bearded fireworm, is a large predator/scavenger polychaeta, present in warm and temperate areas of the Caribbean Sea, Atlantic Ocean, Red Sea (Fishelson, 1971; Ahrens et al., 2013; Ramos & Schizas, 2023) and the Mediterranean, (Baird, 1868, Simonini et al., 2018; Toso et al., 2022, 2024). Despite native to the Mediterranean, it is also considered as a highly invasive species, due to its increasing expansion.*

Lines 190-191 While some Italian citizen scientists projects are listed, I am missing the inaturalist which includes many records from Sicily.

Thank you for the information, we included the ìNaturalist data in the manuscript.

Line 201: in addition to real-time data collected from maritime users. Who are the maritime users? Please explain

Thanks for the comment we integrated this into the manuscript and the text changed as follows:

*The questionnaire built upon the observations by Righi et al. (2020) was administered in 2023 to a diverse group of individuals, yielding a total of 151 responses. Distributed primarily through websites, such as the "Monitoraggio del Vermocane", and the "Fauna Marina Mediterranea" (https://www.facebook.com/groups/230601830399549) and via social media pages of various authors, it was also shared with Italian scuba diving centers. The questionnaire comprised 19 questions, of which four single-choice questions, focusing on the frequency of sightings, the abundance of specimens, and perceptions of the species as a potential issue, were analyzed in this study. Most respondents reported observing the fireworm during scuba diving (74%), spearfishing (10%), and snorkeling (15%) activities, while no questions about the respondents' age or employment status were included. Due to its distinct*

*morphology and vibrant coloration, H. carunculata is unlikely to be mistaken for other species, as no comparable organisms exist in the Mediterranean*

**Results**

Figure 3. A very heterogeneous figure. Better split the productions and oceanographic data from those of the questionnaire.

Thank you for the comment, which is absolutely agreed with, however regarding the standard of the journal there is a limited number of figures and therefore it is necessary to keep the figure with the oceanographic part and the questionnaire.

It is not clear in the methodology how the number of observations has been calculated. Have you consulted other sources than the ones you refer to in methodology? Inaturalist?

Thank you for your comment, we included 'Naturalist and provided more information about the questionnaire. The results from ìNaturalist were highlighted in Figure 3 (green cross)

**Discussion**

If discards is taken as a proxy for damage on fisheries it should be stated clearly. As it stands, in the discussion elaborates on issues that are not visible in the results.

Thanks for the comment, we have clearly defined this concept:

*Recent studies in the area have shown that C. sapidus preys on bivalve mollusks, as evidenced by claw marks on shells. Compared to previous years, mussel (Manila clam) production in 2023 declined by 75 % in the Scardovari lagoon and 100 % in the Canarin lagoon (Azzurro et al., 2025; Chiesa et al., 2025, Tiralongo et al., 2025), for these reason we take the discard data as proxy of C. sapidus biomass and damage on fisheries.*

E.g. Line 311: The proliferation of the blue crab in the  Po river delta has significantly disrupted the clams production (Figure S4). Figure S4 should be presented and commended in the results section.

Thanks for the suggestion we referred to these figure in the discussion section:

*Furthermore, releasing egg-filled C. sapidus females into lagoons (where they cannot be sold on the market, as they don't seem appetizing, Figure S4) can significantly increase crab larvae population in the lagoon. These examples underscore how insufficient management practices can unintentionally enhance the spread of invasive species.*

*This notable increase in the monthly biomass of the C. sapidus in the Northern Adriatic and the annual records of the H. carunculata along the coasts of Sicily can have various implications. Specifically, we show that the proliferation of C. sapidus in the Po river delta has significantly disrupted clam production in 2023 (Figure S4) and under current conditions, no resurgence of clams populations is expected in 2024. The damage to the clam industry is further exacerbated by the costs of removing crabs discarded by fisheries (Figure S4).*

Lines 373-374: Preliminary results show  that H. carunculata consumes mussel meat at high rates, leaving the shells almost completely clean. Results of this study? If not please cite the source of info.

This is a preliminary result of a specific study performed in Simonini et al 2024, we moved the reference in a better position.

*Recent biorefinery research explores its use in processing mollusk waste, specifically expired mussels from retailers, for shell recovery and valorization. Preliminary findings show that H. carunculata consumes mussel meat at high rates, leaving the shells almost completely clean (Simonini et al., 2024). Given its ability to maintain them at high densities without substrate, the species could prove useful for valorizing waste shells, with clean shells serving as a source of "green" calcium carbonate (Seesanong et al., 2023). Developing practical applications for this invasive species could also support removal interventions from areas where H. carunculata is becoming a pest (Simonini et al. 2024).*

Sorry for this and thanks, we will change it.

---

## Author Response (AR2)

Dear Editor,

We would like to thank you and the reviewer for your constructive comments and favourable assessment of our revised manuscript.

Following the suggestions, we have carefully revised the manuscript to improve its grammar and overall readability. We have also restructured and expanded the Discussion and Conclusions sections to better contextualise our findings in the current scientific literature. In doing so, we have included several new references to support and justify our findings.

To improve clarity, we have also split the formerly single Discussion and Conclusions section into two separate paragraphs.

Please note that the new considerations added to the text are highlighted in red and the newly included references in the reference list are also highlighted in red.

We hope that the revised version meets your expectations, and we remain available for any further clarification.

Kind regards,

Riccardo Martellucci